# Discovery of Small Molecules That Inhibit MYC mRNA Translation Through hnRNPK and Induction of Stress Granule-Mediated mRNA Relocalization

**DOI:** 10.3390/ijms26178139

**Published:** 2025-08-22

**Authors:** Yoni Sheinberger, Rina Wassermann, Jasmine Khier, Ephrem Kassa, Linoy Vaturi, Naama Slonim, Artem Tverskoi, Aviad Mandaby, Alik Demishtein, Mordehay Klepfish, Inbal Shapira-Lots, Iris Alroy

**Affiliations:** Anima Biotech, 2 Shoham St., Ramat Gan 5251003, Israel; yoni.sheinberger@animabiotech.com (Y.S.); moty.klepfish@animabiotech.com (M.K.);

**Keywords:** *MYC*, selective translation regulation, RNA-binding protein complex, stress-granules, cancer

## Abstract

*MYC* is a key oncogenic driver frequently overexpressed in non-small cell lung carcinoma (NSCLC) and other cancers, where its protein levels often exceed what would be expected from *MYC* mRNA levels alone, suggesting post-transcriptional regulation. Strategies to inhibit *MYC* function by targeting mRNA translation hold potential for therapeutics utility in Myc-dependent cancers. We developed TranslationLight, a high-content imaging platform which detects *MYC* mRNA translation in human cells. Using this system, we conducted a high-throughput screen of ~100,000 compounds to identify small molecules that selectively modulate *MYC* translation. Candidate compounds were evaluated by immunofluorescence, ribosome profiling, RNA sequencing, cellular thermal shift assays (CETSA), and subcellular localization studies of mRNA and RNA-binding proteins. We identified a lead compound, CMP76, that potently reduces Myc protein without substantially decreasing its mRNA abundance. Mechanistic investigations showed that the compound induces relocalization of *MYC* mRNA into stress granules, accompanied by translational silencing. CETSA identified hnRNPK as a primary protein target, and compound treatment triggered its cytoplasmic relocalization together with formation of hnRNPK-containing granules colocalizing with *MYC* mRNA. Analysis across cancer cell lines revealed that sensitivity to CMP76 was significantly associated with RBM42 dependency. This work establishes a novel therapeutic strategy to inhibit *MYC* translation mediated by hnRNPK, offering a translationally targeted approach to cancer therapy.

## 1. Introduction

Cancer remains the leading cause of mortality worldwide, with non-small cell lung carcinoma (NSCLC) representing approximately 85% of all lung cancer diagnoses. Despite advances in precision oncology and immunotherapy, the prognosis for advanced NSCLC remains poor, necessitating novel therapeutic strategies that extend beyond traditional genotoxic or receptor-targeted approaches.

The transcription factor *MYC* serves as a pivotal driver of oncogenic signaling across multiple cancer types, including NSCLC. As a master regulator of genes controlling cell growth, metabolism, and proliferation, *MYC* overexpression is tightly linked to tumor aggressiveness and therapeutic resistance. Notably, *MYC* is overexpressed in more than 70% of human cancers, including lung, breast, and colorectal malignancies, often correlating with worse clinical outcomes [1,2].

*MYC* mRNA and protein half-life are extremely short; however, the oncogenic potency of Myc is highly correlated with high expression levels of its mRNA and protein [3]. *MYC* is regulated at multiple levels, transcription, mRNA processing, translation, and/or stability mechanisms, potentially involving RNA-binding proteins (RBPs), microRNAs, or stress-induced ribonucleoprotein granules [4].

Stress granules (SGs) are membraneless ribonucleoprotein condensates that form in response to cellular stress, sequestering mRNAs and translational machinery [5]. These dynamic structures function as translational gatekeepers, selectively silencing protein synthesis of pro-proliferative genes like *MYC* during stress responses. Recent work has implicated stress granules in regulating tumor cell survival and chemotherapy resistance, with aberrant mRNA localization into SGs preventing translation of tumor suppressor genes or oncogenes and thereby suppressing or promoting tumor growth under specific conditions [1].

N6-methyladenosine (m6A) represents the most abundant internal mRNA modification in eukaryotic cells and plays crucial roles in mRNA metabolism, including stress granule formation [6]. m6A-modified mRNAs are selectively recruited to stress granules through recognition by reader proteins, suggesting a mechanistic link between RNA modification and translational control during cellular stress [7].

Here, we report the discovery of first-in-class small molecules that modulate *MYC* expression not through transcriptional mechanisms, but by altering the subcellular localization of *MYC* mRNA, relocalizing it to stress granules. Compound-treated cancer cells exhibit stress granule formation and marked reduction in Myc protein despite minimal changes in *MYC* mRNA levels. Our findings establish a novel therapeutic modality targeting translational control via mRNA relocalization, with enhanced efficacy against cancer cell lines exhibiting high dependency on an RNA-binding protein which regulates *MYC* mRNA translation, RBM42 [8].

## 2. Results

### 2.1. Development and Validation of TranslationLight Platform for MYC mRNA Detection

To identify compounds that modulate *MYC* mRNA translation, we employed our high-content imaging platform TranslationLight (TL), previously termed Protein Synthesis Monitoring (PSM) [9]. The TL system utilizes two fluorescently labeled tRNAs (Cy3 and Cy5) that generate a Fluorescence Resonance Energy Transfer (FRET) signal when occupying adjacent ribosomal sites, enabling real-time monitoring of translation at specific codon pairs.

Using codon usage algorithms and translation rate analyses [9], we identified a unique signature codon pair in *MYC* (TCG-CAA, Ser225-Gln226). This codon pair occurs in only 341 of ~20,000 human protein-coding mRNAs, with MYC demonstrating a TL signal-to-noise ratio of 8.8 in A549 cells, indicating 8.8-fold higher ribosomal occupancy at this site compared to other mRNAs containing this codon pair.

To validate *MYC*-selective TL detection, A549 cells were transfected with MYC siRNA or non-targeting control siRNA, followed by transfection with fluorescently labeled tRNAs (Cy3-Gln-UUG and Cy5-Ser-CGU). MYC siRNA treatment resulted in significant reduction in both the calculated FRET (cFRET) signal (Figure 1A,C) and Myc protein levels detected by immunofluorescence (60% reduction, Figure 1B,D). The reduction in TL signal paralleled the decrease in Myc protein expression, confirming the specificity to *MYC* mRNA translation and sensitivity of our detection system.

### 2.2. High-Throughput Screening Identifies Structurally Related MYC Modulators

The validated TL assay was used to screen approximately 100,000 diverse compounds at 30 μM concentration. Compounds were considered hits when their TL scores showed a statistically significant difference compared to DMSO controls [10]. Two structurally related compounds, CMP11 and CMP16, emerged as primary hits with TL scores > 8 (Figure 1E, black circle and square, respectively) [11]. Additional structurally similar but inactive compounds showed TL scores < 4, substantiating structure-activity relationships (Figure 1E, black symbols below the hatched horizontal line).

Hit confirmation experiment validated the screening results, with CMP16 demonstrating superior activity compared to CMP11, consistent with the initial screening data (Figure 1E, gray square and circle, respectively). This prompted further optimization efforts focused on the CMP16 scaffold.

### 2.3. Dose-Dependent Myc Protein Reduction and Anti-Proliferative Activity

Immunofluorescence analysis confirmed dose-dependent reduction in nuclear Myc signal following treatment with hit compounds (Figure 2A,B). CMP16 demonstrated higher potency than CMP11 (EC50 values: 2.4 μM vs. 10.7 μM, respectively), consistent with TL screening results. Structure–activity relationship optimization yielded CMP76 [11], exhibiting significantly enhanced potency (EC50 = 0.24 μM) for Myc protein reduction.

Cell proliferation assays in A549 cells revealed dose-dependent anti-proliferative effects that correlated with Myc inhibition potency, though with consistently lower EC50 values than those observed for protein reduction (Figure 2C), suggesting that reduction in Myc protein is involved in proliferation in A549 cells but may not induce cell death.

Time-course analysis revealed rapid Myc protein depletion, with 20% reduction observed at 2 h, 40% at 6 h, and complete ablation by 24 h (Figure 2E). Western blot analysis confirmed these findings, with CMP16 (30 μM) completely inhibiting Myc protein expression at 24 h while showing modest effects at 2 h (Figure 2D). Puromycilation assay coupled with proximity ligation between anti-puromycin and anti-Myc antibodies was used as an orthogonal assay to TL to support a translation inhibition mode of action of the compounds. CMP16 inhibited puromycin incorporation to Myc nascent peptide, to a similar extent to translation inhibitors, Homoharringtonine and Cycloheximide (Appendix A).

### 2.4. Compound-Induced MYC mRNA Relocalization to Stress Granules

To elucidate the mechanism of Myc protein reduction, we examined *MYC* mRNA localization using fluorescent in situ hybridization (FISH). In control conditions, *MYC* mRNA exhibited homogeneous cytoplasmic distribution as discrete spots representing individual transcripts (Figure 3A upper images, 1 h treatment, cells not labeled by an arrow). Following compound treatment, *MYC* mRNA progressively relocated to larger cytoplasmic granules, with relocalization detectable at 2 h and affecting most cells by 24 h (Figure 3A, upper images, cells marked by arrows). Importantly, this effect was specific to *MYC* mRNA, as GAPDH mRNA distribution remained unchanged.

To determine the nature of these mRNA-containing granules, we examined markers for stress granules (G3BP1) and processing bodies (DCP1a). Compound treatment induced rapid G3BP1-positive stress granule formation beginning at 4 h and increasing in a time-dependent manner through 24 h (Figure 3B and enlarged cell images, Appendix A). Notably, cells containing stress granules were devoid of Myc protein, supporting translational silencing within these structures (Figure 3B and Appendix A). Conversely, DCP1a-positive processing bodies were progressively dissolved, with complete disappearance by 24 h (Figure 3C).

### 2.5. Early m6A-mRNA Relocalization Precedes Stress Granule Formation

Given the established role of m6A modifications in stress granule dynamics, we investigated whether compound treatment affects m6A-mRNA distribution. Immunofluorescence analysis using m6A-specific antibodies revealed that m6A-mRNA relocalization represents the earliest detectable phenotype, occurring within 1 h of treatment (Figure 4). In control cells, m6A signal was uniformly distributed throughout the cytoplasm. Upon compound treatment, m6A-containing granules appeared perinuclearly, becoming larger and more intense through 6 h before dispersing by 24 h. Dose–response analysis revealed an EC50 of 1.2 μM for m6A granule formation, suggesting this represents a primary mechanism of compound action.

### 2.6. RNA-Sequencing Reveals Selective mRNA Stabilization with Shared Regulatory Motifs

To assess compound selectivity and identify additional target mRNAs, we performed RNA-sequencing analysis at 1 h post-treatment, coinciding with m6A-mRNA relocalization. Treatment with active compound CMP16, but not structurally similar inactive control, resulted in significant upregulation of 31 mRNAs (p.adj < 0.05, >50% increase) (Figure 5A).

Motif enrichment analysis using ENCORE eCLIP [12] data revealed that upregulated mRNAs were significantly enriched for pyrimidine-rich (21/31 mRNAs) and AU-rich (18/31 mRNAs) motifs in their 3′-UTRs, with 13 mRNAs containing both motifs (Figure 5B). Analysis of RNA-binding protein interactions showed that 6/31 genes are bound by stress granule constituents (G3BP1, TIA, TIAL1, EIF4G2, ELAVL1), 9/31 genes by processing body components (DDX6, PUM1, PUM2, UPF1), and 11/31 genes by m6A-dependent RBPs (IGF2BP1, IGF2BP2, RBM15). To assess whether specific RBPs preferentially target this gene set, we performed enrichment analysis comparing the frequency of RBP-binding sites in the 3′-UTRs of our 31 upregulated genes versus the genome-wide distribution of these binding sites across all human transcripts. Enrichment analysis revealed that ten RBPs demonstrated 2–5 fold higher binding site density within the 3′-UTRs of compound-responsive genes compared to the background frequency observed across the human transcriptome, suggesting that these may be relevant to the compounds’ mode of action (Figure 5D). The enriched RBPs play a part in p-body formation (DDX6), stress granule formation (TIA and TIAL1), or bind to m6A-modified mRNAs (RBM15) (Figure 5D). FUBP3, far upstream binding protein 3, is a DNA- and RNA-binding protein which belongs to a family of three proteins, FUBP1-3 [13]. FUBP1 collaborates with TFIIH and additional transcription factors for optimal transcription of the *MYC* gene. The three members have a consensus in RNA-binding domain, KH domain, and were reported to bind to AU-rich motifs found in the 3′-UTRs of target mRNAs and to promote their translation. Thus, all enriched RBPs are involved in translation regulation of target mRNAs.

### 2.7. hnRNPK Identified as Primary Compound Target Through CETSA Analysis

To identify protein targets for the compound, we performed cellular thermal shift assay (CETSA) using A549 cell lysates treated with active compound or structurally related inactive control. The CETSA method identifies proteins whose thermal stability is altered upon compound binding, which can reflect either direct engagement of the compound with the protein itself or indirect effects arising from disruption of protein complexes or protein–protein interactions.

Six proteins showed robust, specific thermal stability changes upon active compound treatment (amplitude > 0.2, p.adj < 0.05) (Figure 6). Among the proteins identified, hnRNPK was the only RNA-binding protein detected. Although hnRNPK was found to bind one of the mRNAs upregulated in our RNA-seq analysis, it did not show widespread enrichment across the target transcripts (Figure 5C). This observation is notable given that hnRNPK is known to undergo extensive post-translational modifications, which can dynamically alter both its RNA-binding affinity and its subcellular distribution [14,15]. Importantly, the RBP–mRNA interaction data presented in Figure 5C are derived from pulldown experiments performed in HepG2 and K562 cells. These cell types may differ from A549 cells in their hnRNPK modification status. Moreover, it is conceivable that treatment with the compound induces additional post-translational modifications of hnRNPK, potentially explaining the lack of correlation between hnRNPK binding and the stabilized mRNAs observed in our study.

Additional proteins showing significant but lower-magnitude effects included other mRNA regulatory proteins: RBPs (GTF2F1, YBX3), translation factors (EIF2B3, HSPA14), mRNA-modifying enzymes (NAT10, XRN2), and nuclear export factors (NUP93, NUP205, XPO7). The convergence of these findings suggests compound-mediated disruption of RNA regulation pathways, potentially through interference with hnRNPK-containing ribonucleoprotein complexes. For example, RBM42 is an RBP which regulate *MYC* mRNA translation, and is an hnRNPK-interacting protein [15,16]. Although it was not identified herein, eleven of CMP76 CETSA targets are also RBM42-interacting proteins, namely, MAP7D3, EIF2B3, NAT10, YBX3, NUP205, TK1, PPP1R12A, XRN2, CCDC86, NUP93, and hnRNPK.

### 2.8. Ribosome Profiling Confirms Translational Repression Despite mRNA Stabilization

To distinguish between effects on mRNA stability versus transcription, we performed ribosome profiling following 6 h of CMP16 treatment. Analysis revealed stabilization of 87 mRNAs (>2-fold increase, p.adj < 0.05), with 28 of these genes representing known hnRNPK targets based on ENCORE eCLIP data [12] (Figure 7A).

Despite mRNA accumulation, translation efficiency (ribosome footprint/mRNA ratio) was decreased, indicating that the observed increase in mRNA expression results from reduced degradation of translationally silenced transcripts rather than enhanced transcription (Figure 7B). This supports our model wherein compounds promote mRNA sequestration in stress granules, protecting transcripts from degradation while blocking translation.

Analysis of downregulated genes revealed enrichment for *MYC* transcriptional targets, consistent with the observed 30% Myc protein reduction at 6 h and supporting downstream transcriptional consequences of Myc depletion.

### 2.9. hnRNPK Subcellular Relocalization Accompanies Compound Treatment

hnRNPK is highly post-translationally modified by phosphorylation, arginine methylation and K63-ubiquitination which regulate its function, cellular localization and mRNA binding [14,15]. Thus, we examined hnRNPK subcellular localization following 24 h compound treatment. Combined immunofluorescence and RNA FISH analysis revealed that compound treatment induced hnRNPK relocalization from predominantly nuclear to cytoplasmic distribution, with formation of cytoplasmic hnRNPK-containing granules in some cells (Figure 8). These granules colocalized with stress granule marker FXR1 and contained *MYC* mRNA, supporting a model wherein compound treatment disrupts normal hnRNPK function and promotes its sequestration in stress granules along with target mRNAs.

We have attempted to identify which modification drives the compound effect on hnRNPK relocalization. However, we could not detect short term (<2 h) changes in two sites of hnRNPK phosphorylation (Ser216 and ser286) nor in its methylation). Thus, it may be that the change in localization is due to post-translation modifications we did not test, such as ubiquitination, or due to disruption of hnRNPK complex formation which results in its translocation.

### 2.10. Cancer Cell Line Sensitivity to CMP76 Are Dependent on RBM42 Expression

To assess therapeutic potential, we evaluated compound activity across a panel of 43 cancer cell lines. We have analyzed the correlation between compound potency and *MYC*, *HNRNPK* and *RBM42* expression levels (mRNA and protein) and dependency (CRISPR and siRNA) [17,18]. Analysis of cells which are highly sensitive to the compound (pEC50 > 5.5, hill slope < −3.5, Figure 9A) versus cell lines which show low sensitivity to the compound (Figure 9B) demonstrated a statistically significant *RBM42* dependency (*p*-value = 0.0432), though with some data dispersion suggesting additional complexity which may be driven by other proteins in the complex or their regulation (Figure 9C).

## 3. Discussion

This study uncovers a previously unrecognized mechanism for *MYC* regulation through small molecule–induced relocalization of *MYC* mRNA to stress granules. In contrast to conventional approaches that target *MYC* transcription or protein stability, the compounds identified in this work act at the post-transcriptional level, altering the subcellular localization of *MYC* mRNA and effectively sequestering it within translationally repressive compartments. This relocalization leads to rapid depletion of Myc protein, detectable as early as two hours post-treatment—a timeframe consistent with the protein’s short half-life (~1 h in A549 cells). These kinetics suggest that translational inhibition, rather than transcriptional suppression, is the principal driver of protein depletion, offering a potentially advantageous therapeutic profile through swift modulation of oncogene expression.

Temporal dissection of the compound’s activity indicates that the earliest detectable cellular phenotype is a redistribution of m6A-modified mRNAs, occurring within one hour of treatment and preceding overt stress granule formation. This observation implicates perturbation of m6A-RNA regulatory networks as an initiating event, with stress granule assembly emerging as a downstream consequence. The selective recruitment of m6A-tagged transcripts into stress granules is consistent with published data showing that m6A modified mRNAs promote granule formation via interactions with reader proteins [19]. Moreover, several RBPs identified as binding to compound-responsive mRNAs—such as IGF2BP1, IGF2BP2, and RBM15—are themselves m6A-binding proteins, further supporting this mechanism [20,21]. These RBPs have been shown to play pro-proliferative roles in lung adenocarcinomas and NSCLC and are upregulated in those contexts [20], highlighting the clinical relevance of this regulatory axis.

Mechanistically, the identification of hnRNPK as a principal compound target provides critical insight into the observed phenotypes. hnRNPK is a multifunctional RNA-binding protein that contributes to diverse aspects of RNA metabolism, including transcription, alternative splicing, mRNA export, and translation [14,22]. Upon compound treatment, hnRNPK translocates from the nucleus to the cytoplasm, a change that may be driven by altered post-translational modifications or disruption of RNA-binding protein complexes. Given that hnRNPK participates in large ribonucleoprotein complexes, it is likely that the compound disrupts these assemblies, selectively affecting the translation of specific mRNAs [14,23]. Interestingly, only a subset of hnRNPK-associated transcripts were stabilized by the compound, suggesting selective modulation of hnRNPK complexes based on RNA motif preferences or context-dependent complex composition. This selectivity may be shaped by specific post-translational modifications; for example, K63-linked ubiquitination of hnRNPK restricts *MYC* translation, and loss of the E3 ligase Fbxo4 enhances *MYC* expression by allowing unregulated hnRNPK activity [15].

Compound sensitivity across cancer cell lines correlates with dependency on the RBP *RBM42*, which has been shown to bind to the *MYC* 5′ UTR and enhance its translation. The compound may interfere with this interaction, promoting *MYC* mRNA sequestration into stress granules. Notably, hnRNPK is known to bind to RBM42 and modulate cellular metabolism [16], further suggesting that disruption of hnRNPK–RBM42 interactions could underpin compound activity. Future work focused on dissecting the specific modifications or protein–protein interactions required to elicit this effect may yield biomarkers of compound responsiveness and support the development of more potent analogs.

Stress granules (SGs) are membraneless organelles that orchestrate RNA localization and translational control [24]. Once *MYC* mRNA is sequestered into SGs, translation is inhibited primarily through spatial exclusion of the mRNA from active translating ribosomes. Stress granules are enriched in stalled translation initiation complexes, RNA-binding proteins, and translational repressors, but lack the complete ribosomal machinery required for elongation [24]. Within these condensates, mRNAs are physically separated from ribosomes and translation initiation factors, preventing ribosome loading and elongation. Although there is in depth understanding of the mechanism of SGs formation, the protein and mRNAs that constitute it, the exact functional role is unclear [25]. hnRNPK relocalization to the cytoplasm may reinforce *MYC* mRNA translation repression by stabilizing it in a translationally silent state—either through its interaction with *MYC* 5′-UTR thereby competing with translation initiation factors or by modulating associated RBPs, such as RBM42. This compartmentalization effectively shifts *MYC* transcripts from the actively translating pool into storage-prone state, silencing protein synthesis until granule disassembly.

Beyond *MYC*, these findings establish proof-of-concept for a broader therapeutic paradigm: targeting mRNA localization and translation via small molecules. The TranslationLight platform enables systematic identification of compounds that influence codon-specific translation, revealing mRNA-specific vulnerabilities that are not apparent through conventional transcriptomic or proteomic screening. This approach could be readily extended to other oncogenes with similarly complex post-transcriptional regulations.

To translate these findings toward clinical application, future work should evaluate compound activity in in vivo models and characterize potential toxicities. Combination strategies with existing therapies could enhance efficacy or expand indications, particularly in malignancies with known *MYC* dependencies. The high sensitivity observed in lymphoma cell lines suggests a potential early path for development in hematologic cancers, where post-transcriptional regulation of Myc is often critical for pathogenesis.

In conclusion, we have discovered and characterized a new class of small molecules that suppress Myc protein expression through the relocalization of its mRNA to stress granules. This work introduces RNA subcellular localization as a therapeutically tractable axis of gene regulation, with potential applications beyond *MYC*. The identification of hnRNPK as a primary target and the link between compound sensitivity and RBM42 dependency highlight opportunities for biomarker-driven patient selection.

## 4. Materials and Methods

Antibodies

Rabbit anti-Myc antibody (Abcam Limited, Cambridge, UK, ab32072); Rabbit anti-GAPDH antibody (Abcam Limited, Cambridge, UK, ab37168); Mouse anti-G3BP1 (Abcam Limited, Cambridge, UK, ab56574); Rabbit anti-DCP1A (Abcam Limited, Cambridge, UK, ab47811); Rabbit anti-FXR1 antibody (Abcam Limited, Cambridge, UK, ab129089); Mouse anti- N6-methyladenosine (m6A) antibody (Abcam Limited, Cambridge, UK, ab208577); Mouse anti-HNRNPK antibody (Abcam Limited, Cambridge, UK, ab70492); Mouse anti-puromycin antibody (Merck KGaA, Darmstadt, Germany, MABE343); Mouse anti-Tubulin beta1 antibody (Abcam Limited, Cambridge, UK, ab204947); Donkey Anti-Rabbit IgG H&L (Alexa Fluor^®^ 647) (Abcam Limited, Cambridge, UK, ab150075); Donkey Anti-Mouse IgG H&L (Alexa Fluor^®^ 488) (Abcam Limited, Cambridge, UK, ab150105).

Cell culture

A549 cells (ATCC^®^, Manassas, VA, USA, CCL-185™) were maintained in DMEM low glucose medium (IMBH, Beit Haemek, Israel, Cat. 01-050-1A), containing 10% fetal bovine serum, 1% L-glutamine, and 1% penicillin-streptomycin solution.

TranslationLight screen and analysis

Cy3 and Cy5 Labeled tRNA [26] were transfected with 0.4 µL HiPerFect (Qiagen, Hilden, Germany) per 384 well. First, HiPerFect was mixed with DMEM and incubated for 5 min; next, 6 nanograms Cy3- labeled tRNAgln and 6 ng Cy5-labled tRNAser were diluted in 1xPBS and then added to the HiPerFect:DMEM cocktail and incubated at room temperature for 10 min. The transfection mixture was dispersed automatically into 384-well black plates. Cells were then seeded at 3500 cells per well in complete culture medium and incubated at 37 °C, 5% CO2. Forty-eight hours after transfection compounds (100K ChemDiv diversity set, G-INCPM, Weizmann Institute, Rehovot, Israel) were added at a final concentration of 30 µM. Four hours post-treatment, cells were fixed with 4% paraformaldehyde and images were captured with Operetta microscope (Revvity, Waltham, MA, USA) using ×20 high NA objective lens. Quantitative Fluorescent Energy Transfer between two fluorophores, Cy3 and Cy5 in this article, requires correction (cFRET) due to bleedthrough between fluorescence channels [27]. We have added the cFRET calculation during image upload process to the cloud, generating a fifth channel that is used for TL assessment algorithms.

The TranslationLight analysis platform extracts both direct and derived imaging features (10 features) that capture translation dynamics at the cellular level. One such feature, the Normalized cFRET, serves as a proxy for FRET efficiency, quantifying the proportion of energy transferred from donor to acceptor fluorophores. This efficiency measure is based on established FRET theory [28] and provides a robust, normalized readout of translation activity. Integrating diverse readouts into a single hit score is nontrivial, as inter-feature dependencies can introduce bias. To address this, we employ a multi-dimensional statistical framework that assigns a *p*-value to each compound based on its deviation from a reference cell population (H_0_). This *p*-value-based approach enables aggregation of multiple features into a single, unified significance score (namely aggregated *p*-value) while appropriately accounting for inter-feature correlations [10].

In parallel, we also calculate a Screen score, which quantifies the deviation of a compound’s feature profile from the reference population (H_0_) in units of standard deviations. This normalized score reflects the magnitude of perturbation, with larger absolute values indicating stronger divergence from the null distribution. While the aggregated *p*-value emphasizes statistical significance, the Screen score provides an interpretable effect size for compound activity across the multi-parametric readouts.

Proximity ligation coupled to puromycilation assay

For visualization of Myc protein synthesis, cells were subjected to a puromycylation assay (SUnSET) adapted for immunofluorescence. Briefly, cells were seeded on in 384-well plates and treated with compounds or DMSO for 1 h, followed by addition of puromycin (final concentration 10 µg/mL) directly to the culture medium for 10 min at 37 °C to label nascent polypeptides. In negative control wells, puromycin was not added. After labeling, cells were rapidly rinsed with warm PBS, fixed with 4% paraformaldehyde in PBS for 10 min at room temperature, and permeabilized with 0.1% Triton X-100 in PBS for 10 min. Following blocking in Duolink^®^ Blocking Solution for 60 min, cells were incubated with mouse anti-puromycin antibody (e.g., clone 12D10, 1:1000 in blocking buffer) and rabbit anti-Myc antibody (1:1000) for 1.5 h at room temperature, then washed three times in PBS. Proximity ligation was performed with anti-rabbit and anti-mouse Duolink^®^ antibodies, according to the manufacturer protocol (Merck KGaA, Darmstadt, Germany). Nuclei were counterstained with DAPI, and cytoplasm with a mouse anti-tubulin beta 1 antibody which was visualized with a secondary Goat-anti-mouse Alexa Fluor 647. Images were acquired using identical exposure settings across conditions.

Immunofluorescence assays

A549 cells were grown in 384-well plates (Cat. 6057300, Revvity, Waltham, MA, USA) for 48 h, treated with compounds for the indicated time and then fixed for 20 min in 4% paraformaldehyde. Permeabilization was performed using 0.1% Triton X-100 in 4%FBS and PBS for 5 min. Primary antibody staining was performed for 90 min at room temperature or overnight. Cells were then washed twice with PBS and incubated with a secondary antibody for 90 min at room temperature. Nuclei were stained with DAPI for 10 min and washed twice with PBS. Cell images were taken with Operetta (Revvity, Waltham, MA, USA), a wide-field fluorescence microscope at 20x magnification. After acquisition, the images were transferred to AWS cloud and Columbus software (Revitty, Waltham, MA, USA) for image analysis. In Columbus, cells were identified by their nucleus, using the “Find Nuceli” module and cytoplasm was detected using the digital phase channel. Subsequently, the fluorescent signal was enumerated in the identified cytoplasmic, granules, or nuclei region. Then, data were exported to a data analysis and visualization software (Tibco Spotfire^®^ Analyst, v.12.0.3, USA).

Fluorescent In Situ Hybridization (FISH) assay

A549 cells were grown in 384-well plates (Cat. 6057300, Revvity, Waltham, MA, USA) for 48 h, treated with compounds for 4 h and then fixed for 20 min in 4% paraformaldehyde. Next day, permeabilization was performed for 90 min at 4 °C, using 70% ethanol. Then, the cells were incubated for 10 min with 10% formamide in 10% saline–sodium citrate. Fluorescently labeled custom DNA probes that target *MYC* (Cy3, BioSearch Technologies, Petaluma, CA, USA, Cat. SMF-1063-5) and GAPDH (Cy5, BioSearch Technologies, Petaluma, CA, USA, Cat. SMF-2019-1) mRNAs were hybridized overnight at 37 °C in a dark chamber in 10% formamide. The next day, cells were washed twice with 10% formamide for 30 min. Next, nuclei were counterstained with DAPI (Merck KGaA, Darmstadt, Germany, Cat. 5MG-D9542) and then cells were washed twice with PBS. FISH experiments were performed according to the probes manufacturer’s protocol for adherent cells. Following RNA FISH experiments, images of cells were taken with Operetta (Revvity, Waltham, MA, USA), a wide-field fluorescence microscope at x63 magnification. After acquisition, the images were transferred to AWS cloud and Columbus software for image analysis. In Columbus, cells were identified by their nucleus, using the “Find Nuceli” module, cytoplasm was detected based on the FISH channel or digital phase, and single mRNAs in the cytoplasm and transcription sites in the nucleus were detected using “Find Spots” module. Subsequently, fluorescent signals were collected for each channel in the identified regions: nucleus, cytoplasm, and spots. Data were exported to a data analysis and visualization software (Tibco Spotfire^®^ Analyst, v.12.0.3, USA).

Image Acquisition and Analysis

High-throughput image acquisition was conducted using the Operetta^®^ CLS™ microplate imaging system (Revvity, Waltham, MA, USA). Image analysis was carried out with the Columbus™ Image Data Storage and Analysis platform (Revvity, Waltham, MA, USA), which facilitates cell segmentation and the extraction of cytological features—such as nuclear and cytoplasmic morphology—as well as quantification of intensity across defined regions of interest (e.g., whole cell, cytoplasm, perinuclear region, spots, nucleus). The resulting single-cell data were further processed through our proprietary analytical pipelines, which include machine learning and artificial intelligence (AI) approaches—some of which operate directly on the raw image data without reliance on Columbus preprocessing—as well as TranslationLight-specific statistical frameworks. The analysis pipeline is designed for large-scale parallelization in a cloud environment, enabling the execution of hundreds of instances simultaneously and supporting rapid turnaround for entire screening campaigns. The data are extracted and exported into a visualization software, Spotfire (Tibco Spotfire^®^ Analyst, v.12.0.3, USA).

Comparison circles were used to visualize statistical differences between group means in data derived from cell-level data in imaging experiments. Each circle is centered on the group mean, with a radius determined by the Tukey–Kramer method to reflect standard error and multiple comparison adjustments. Non-overlapping circles indicate statistically significant differences between means, while overlapping circles suggest non-significant differences [29].

RNA Sequencing [Massive Analysis of cDNA Ends (MACEseq)]

A549 human lung carcinoma cells were seeded in 6 cm tissue culture dishes (400 × 10^3^ cells per dish) in growth medium consisting of low glucose DMEM supplemented with 10% fetal bovine serum (FBS). After 48 h of incubation (approximately 70% confluence), cells were treated with compound (10 µM) or DMSO (0.1%, vehicle control) for 1 h in triplicates.

Following treatment, cells were washed twice using 1× PBS (AM9625, Thermo Fisher Scientific, Waltham, MA, USA), at room temperature. The PBS was aspirated thoroughly, and cells were scraped in 350 μL RLT buffer (79216 Qiagen), supplemented with 1% β-Mercaptoethanol (M3148 Merck KGaA, Darmstadt, Germany). The resulting cell pellet was collected into 1.5 mL tube, flash frozen with liquid nitrogen, stored at −80 °C, and shipped on dry ice. Libraries for MACEseq were prepared by GenexPro using the MACE Kit (GenXPro GmbH, Frankfurt, Germany). Briefly, the library preparation starts with extraction of total RNA, followed by polyadenylated mRNA enrichment using the Dynabeads mRNA Purification Kit (Life Technologies, Waltham, MA, USA). cDNA is prepared via reverse transcription with biotinylated oligo (dT) primers and then fragmented to an average size of 250 bp by sonication with a Bioruptor (Diagenode, Seraing, Belgium). Streptavidin beads were used to capture the biotinylated cDNA 3′ ends, which were subsequently tagged with TrueQuant DNA adapters (Waltham, MA, USA). The libraries are then PCR amplified, purified using SPRI beads (Agencourt AMPure XP; Beckman Coulter, Brea, CA, USA), and sequenced on a NextSeq 500 (Illumina Inc., San Diego, CA, USA). Bioinformatic analysis was performed by using GenXPro’s proprietary differential expression and alternative polyadenylation (APA) analysis pipeline.

To identify the RBPs which bind differentially upregulated mRNAs (DE), we downloaded eCLIP binding data, 5 January 2025 [12]; this dataset includes binding sites for 168 RBPs in two cell lines (accession numbers and files listed in Appendix A). For each DE gene, we found the RBPs bound to it (based on GENCODE annotation v.44) [30], and classified each binding site as 5′UTR, CDS, 3′UTR, or intron; 115 RBPs overall were identified this way. We then classified each of the 115 RBPs, using GO:BP (gene ontology: biological process) terms [31,32], into one or more of the following categories: involved in splicing; involved in P-bodies assembly; and involved in assembly or disassembly of stress granules. The GO terms used for this categorization appear in Appendix A. Categorization of RBPs belonging to stress granules or P-bodies was performed using ChatGPT v.4.0 [33] and verified by us for the RBPs identified herein.

Cellular Thermal Shift Assay (CETSA)

A549 human lung carcinoma cells were cultured in T-175 tissue culture flasks under standard conditions (37 °C, 5% CO_2_) until reaching approximately 80–90% confluency, yielding a total of 8 × 10^7^ cells. Cells were detached using 0.25% Trypsin-EDTA (Thermo Fisher Scientific, Waltham, MA, USA, Cat. No. 25200-072) and collected by centrifugation at 500× *g* for 5 min at room temperature. The cell pellet was washed once with Hanks′ Balanced Salt Solution (HBSS) (Merck KGaA, Darmstadt, Germany, Cat. No. 55037C), followed by a second centrifugation at 500× *g* for 5 min. The final cell pellets were immediately shipped on dry ice to Pelago Bioscience AB (Solna, Sweden) for CETSA^®^ analysis under standardized assay conditions. CETSA analysis was performed according to the general CETSA^®^ protocol [34] with detection by the liquid chromatography–mass spectrometry (LC-MS) method and with implementation of a seven concentration CETSA^®^ MS profiling strategy in the compressed format. The study included protein stability assessment in cells treated for 15 min at 25 °C with the compounds at seven different concentrations (0.1 µM, 0.3 µM, 1 µM, 3 µM, 10 µM, 30 µM, and 100 µM) relative to non-treated control (0.1% DMSO). Treated cells were aliquoted and subjected to a heat challenge at 12 different temperatures (44–66 °C), after which individual samples from all temperature points were pooled for each test condition. Aggregated proteins were removed using centrifugation and soluble protein amounts were measured using LC-MS. The individual protein concentration response curves were analyzed and molecular targets were identified when response to the compound was observed. Further, unheated samples treated with compounds at highest concentration (100 μM) as well as vehicle control (DMSO) were analyzed to further differentiate compound-induced protein thermal stability and protein abundance alterations. The experiment was performed for each compound in two replicates.

Ribosome Profiling (Ribo-Seq)

A549 human lung carcinoma cells were seeded in 15 cm tissue culture dishes (3 × 10^6^ cells per 15 cm dish) in growth medium consisting of low glucose DMEM supplemented with 10% fetal bovine serum (FBS). For each experimental condition, three plates were pooled to constitute a single biological replicate, with experiments performed in triplicate. After 48 h of incubation (approximately 70% confluence), cells were treated with compound (10 µM) or DMSO (0.1%, vehicle control) for 6 h.

Following treatment, cells were rapidly washed and scraped on ice using 1× PBS (AM9625, Thermo Fisher Scientific, Waltham, MA, USA) supplemented with cycloheximide (100 µg/mL, C4859 Sigma-Aldrich) and collected into 15 mL conical tubes. Ten percent of each pooled lysate was transferred to a separate microcentrifuge tube for subsequent RNA-seq analysis. The remaining lysate was centrifuged at 200× *g* for 5 min at 4 °C. Supernatants were carefully removed, and cell pellets were flash-frozen in liquid nitrogen, stored at −80 °C, and shipped on dry ice to EIRNA BIO (Cork, Ireland).

EIRNA BIO performed quality control, total RNA extraction, ribosome-protected fragment isolation, and library preparation using proprietary protocols. Libraries that passed internal quality control, including evaluation of triplet periodicity, were sequenced on the Illumina Hi Seq X platform. Downstream bioinformatic analysis included raw read processing, transcriptome alignment, and differential assessment of translational efficiency between experimental conditions. For data normalization, read counts were normalized to account for differences in sequencing depth across samples, using the DESeq2 package. The reads were mapped to the human transcriptome (Gencode version 35). Differential gene expression analysis was performed using deltaTE [35], which analyzes differential changes in mRNA counts, ribosome-protected fragments (RPFs), and translational efficiency (TE)—the ratio of RPFs to mRNA counts—for each gene, also employing the DESeq2 package, v.1.34.

Cancer cell line panel proliferation assay

Cancer cell line panel proliferation assay was conducted by Eurofins Discovery (OncoPanel™ Cell-Based Profiling) (St. Charles, MO, USA). Cells were seeded into 384-well plates at their respective, established cell density in standardized media. At the same time, a time zero non-treated plate is generated. Twenty-four (24) hours later, compound was added, and the time zero plate was developed, for doubling calculations, by cell lysis with cell viability detection reagent (Promega CellTiter-Glo^®^, Promega Corporation, Madison, WI, USA). Test agent plates are incubated continuously for 3 days. Cells were then lysed with cell viability detection reagent (Promega CellTiter-Glo^®^, Promega Corporation, Madison, WI, USA).

Compounds were diluted in DMSO at starting at 30 μM and then serially diluted in DMSO by 3.16-fold to complete the 10-point concentration curves. Compounds were added directly from these dilutions to cell plates using Echo 555 acoustic energy-based transfer.

CMP76 responsive and non-responsive cells were grouped based on their potency and hill slope (responsive, EC50 < 6 μM and hill slope between −1 and −3.5). Statistical analysis was conducted using Welch’s *T*-test (Graphpad, v. 10 Software).

## Figures and Tables

**Figure 1 ijms-26-08139-f001:**
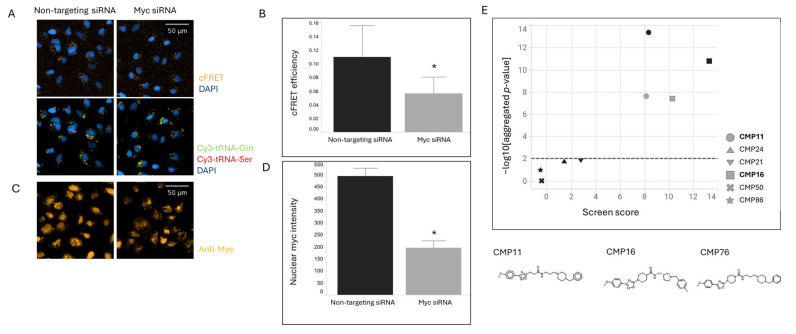
Discovery of compounds modulating *MYC* mRNA regulation using TranslationLight. (**A**) Validation of tRNA pair Gln(UUG)-Ser(CGU) for reporting *MYC* mRNA regulation. A549 cells were transfected with MYC siRNA or non-targeting control siRNA (right and left images, respectively). Twenty-four hours later, fluorescently labeled tRNAs were transfected. Forty-eight hours post-tRNA transfection, cells were fixed, nuclei stained with DAPI, and Cy3, Cy5, and FRET signals captured by fluorescence microscopy (cFRET images top, Cy3 and Cy5 images bottom; orange, green, and red colors, respectively. DAPI is in blue). (**B**) Quantitative analysis of cFRET efficiency following siRNA treatment. Data presented as median cFRET signal normalized to total transfected tRNA per cell for three replicate wells. Black bars: non-targeting siRNA; gray bars: MYC siRNA. Statistical significance (* *p* < 0.05) determined as described in Materials and Methods. (**C**) Validation of Myc protein knockdown by immunofluorescence. Following tRNA imaging, cells were permeabilized with 0.5% Triton X-100 and probed with anti-Myc antibody and Alexa Fluor 647-conjugated secondary antibody. MYC siRNA images (**right**) and non-targeting siRNA (**left**). Top images show anti-Myc staining; lower images show DAPI staining. (**D**) Quantitative analysis of Myc immunofluorescence signal intensity in cell nuclei. Data extracted using Columbus image analysis software v.2.8 and presented as average nuclear Myc signal per cell across three treated wells. Black bars: non-targeting siRNA; gray bars: MYC siRNA. Statistical significance (* *p* < 0.05) determined as described in Materials and Methods. (**E**) Results of *MYC*-selective TranslationLight screen and hit confirmation. Data shown for two structurally related hit compounds in primary screen (CMP11 and CMP16, black circle and square, respectively) and hit confirmation (gray circle and square, respectively). Additional structurally similar compounds that were not identified as hits are also shown. Compounds structures are shown for two hit compounds and an optimized compound, CMP76. X-axis depicts TranslationLight score; Y-axis shows statistical significance relative to DMSO controls. Compounds with scores > 4 and *p*-values < 0.01 (marked by hatched horizontal line) were selected as hits.

**Figure 2 ijms-26-08139-f002:**
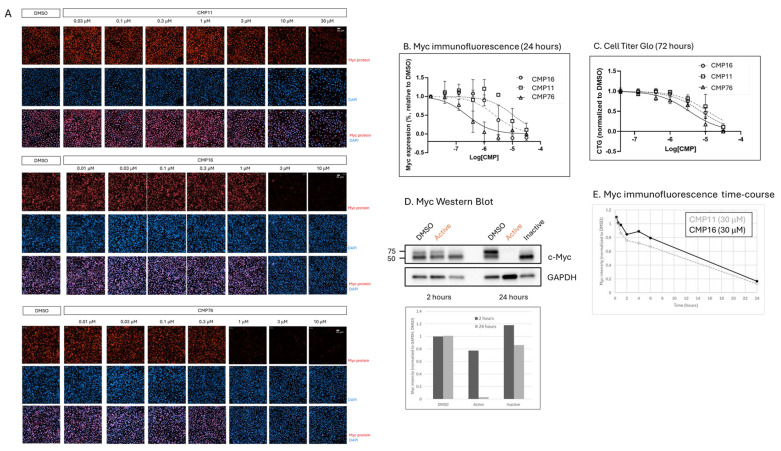
Hit and optimized compounds reduce Myc protein expression and A549 cell proliferation in a dose-dependent manner. (**A**) Dose–response analysis of hit compounds CMP11, CMP16, and optimized compound CMP76 in A549 cells. Cells were treated for 24 h at indicated concentrations, fixed, stained with DAPI and anti-Myc antibody, and visualized by fluorescence microscopy (×20 magnification). Myc staining shown in red; DAPI in blue. (**B**) Dose-dependent reduction in Myc expression quantified by image analysis. Nuclear anti-Myc signal intensity extracted using Columbus software and normalized to DMSO-treated controls. X-axis shows log10-transformed molar compound concentration. EC50 values determined using GraphPad Prism v.10: CMP16 = 2.4 μM, CMP11 = 10.7 μM, CMP76 = 0.24 μM. (**C**) Compound effects on A549 cell proliferation measured by CellTiter-Glo following 72 h treatment. Y-axis shows normalized CTG values relative to DMSO; X-axis shows log10-transformed molar concentration. EC50 values: CMP16 = 7.4 μM, CMP11 = 11.6 μM, CMP76 = 3.5 μM. (**D**) Western blot analysis of Myc protein levels. A549 cells treated with DMSO, CMP16 (30 μM), or structurally similar inactive compound for 2 or 24 h. Equal protein amounts analyzed by SDS-PAGE and immunoblotting with anti-Myc and anti-GAPDH (loading control) antibodies. Band intensities quantified using ImageJ v.1.54 and normalized to GAPDH. Dark bars: 2 h; light gray bars: 24 h. (**E**) Time-course analysis of Myc reduction by CMP11 and CMP16 (30 μM each). A549 cells treated for indicated times, fixed, and stained for Myc immunofluorescence. Nuclear Myc intensity in compound-treated cells normalized to DMSO controls at each timepoint. X-axis: time in hours; Y-axis: normalized Myc intensity.

**Figure 3 ijms-26-08139-f003:**
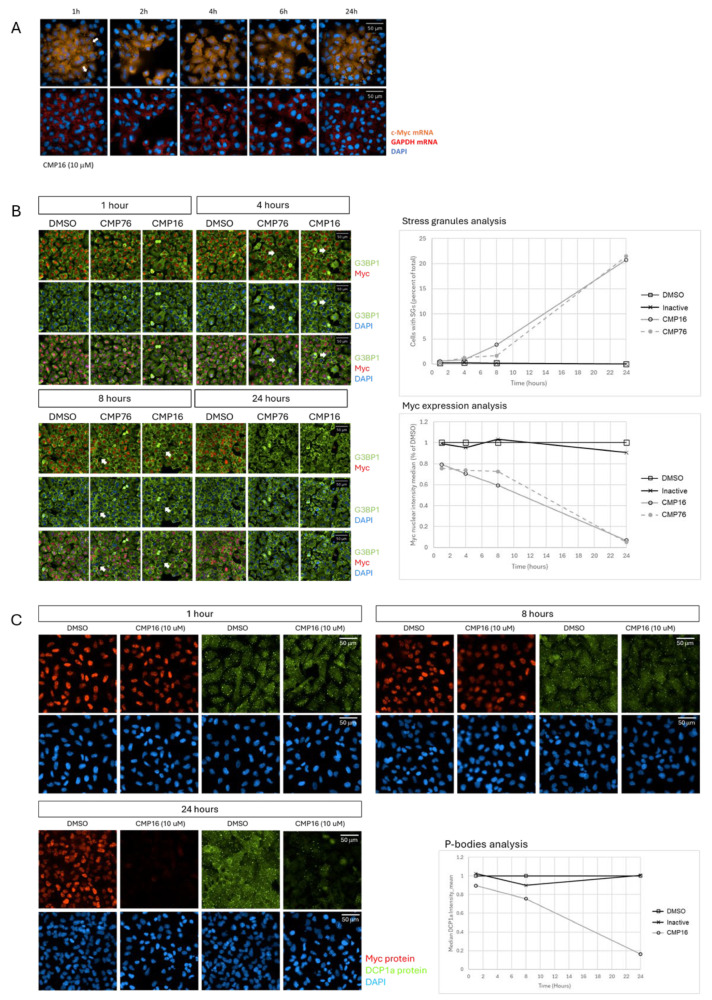
Compounds induce stress granule formation and p-body dispersal. (**A**) *MYC* mRNA relocalization to RNA granules visualized by FISH. A549 cells treated with CMP16 (10 μM) for indicated times, fixed, and processed for FISH using Alexa Fluor-tagged *MYC*-specific probes (**upper panels**) or GAPDH-specific probes (**lower panels**). DAPI staining shows nuclei. White arrows mark cells with *MYC* mRNA granules. (**B**) Stress granule formation detected by G3BP1 immunofluorescence. A549 cells treated with CMP16 or CMP76 for 1, 4, 8, or 24 h, fixed, and stained with anti-G3BP1 and anti-Myc antibodies. Quantitative analysis shows number of cells with stress granules (**upper right**) and nuclear Myc intensity (**lower right**) normalized to DMSO controls versus treatment time. White arrows mark cells with stress granules and devoid of myc expression. (**C**) P-body dispersal detected by DCP1a immunofluorescence. A549 cells treated with CMP16 (10 μM) for indicated times and stained with anti-DCP1a and anti-Myc antibodies. Quantitative analysis shows DCP1a granule intensity normalized to DMSO controls versus treatment time (**lower right panel**).

**Figure 4 ijms-26-08139-f004:**
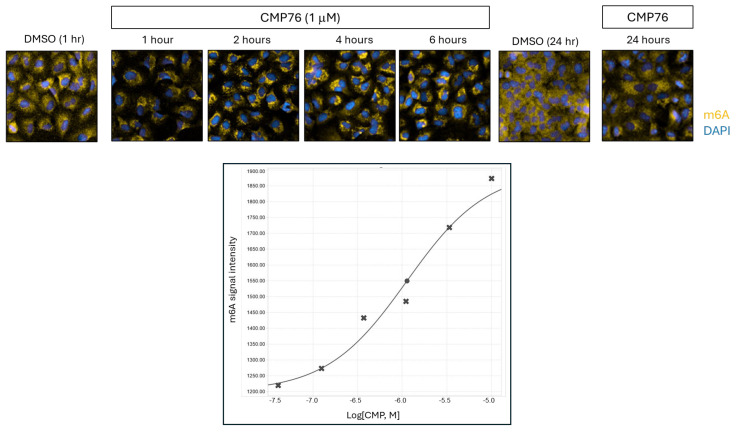
Time-dependent relocalization of m6A-modified mRNAs from diffuse cytoplasmic to perinuclear punctate distribution. A549 cells treated with CMP76 (1 μM) for indicated times, fixed, and stained with anti-m6A antibody. Representative images show progression from uniform cytoplasmic m6A distribution (DMSO) to perinuclear granular pattern. Quantitative analysis shows cytoplasmic m6A signal intensity versus treatment time, fitted with logistic regression curve.

**Figure 5 ijms-26-08139-f005:**
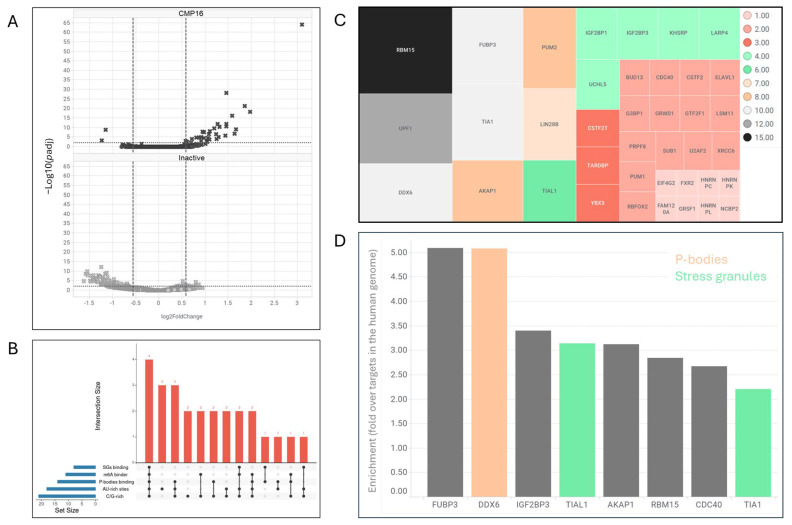
RNA-sequencing analysis reveals selective mRNA stabilization with shared regulatory motifs. A549 cells treated with CMP16 (10 μM) for 1 h and analyzed by RNA-sequencing in triplicate. (**A**) Volcano plot showing log2 fold-change in mRNA expression (X-axis) versus adjusted *p*-value (Y-axis) for compound versus DMSO treatment. Horizontal dashed lines denote statistically significant values, *p* < 0.05; vertical horizontal lines denote fold change above or below 2. (**B**) Analysis of RNA-binding protein interactions with 31 upregulated genes (log2 fold-change > 1, *p* < 0.05) presented as UpSet plot showing overlap between genes containing specific regulatory motifs and RBP-binding sites. (**C**) Treemap visualization of RBPs binding to upregulated genes, with box size and colors representing number of bound genes. (**D**) Enrichment analysis of RBPs enriched for putative binding of compound DE relative to their putative binding sites in the human genome.

**Figure 6 ijms-26-08139-f006:**
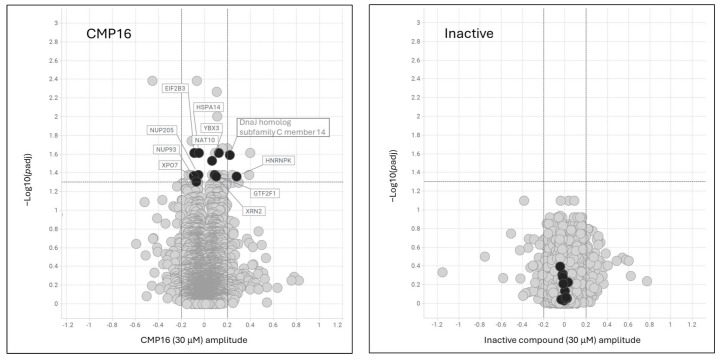
hnRNPK identified as primary compound target by CETSA analysis. Volcano plots showing cellular thermal shift assay results for CMP16 (**left**) versus structurally similar inactive compound (**right**) in A549 cell lysates. X-axis: effect amplitude (log2 fold-change); Y-axis: statistical significance (−log10 *p*-value). Proteins showing significant thermal stability changes upon active compound treatment are labeled by black circles, with hnRNPK identified as the primary RNA-binding protein target. Non-significant or mechanistically non-related proteins are labeled as grey circles.

**Figure 7 ijms-26-08139-f007:**
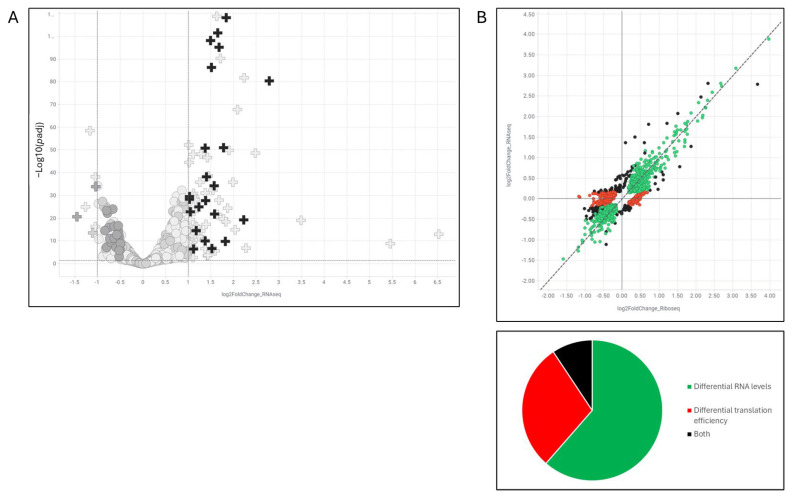
Ribosome profiling reveals increase in mRNA expression and translation repression. A549 cells treated with CMP16 (5 μM) for 6 h and analyzed by RNA-sequencing and ribosome profiling. Black plus signs indicate hnRNPK target genes identified by hnRNPK-eCLIP [12]. Plus signs indicate genes which are statistically significantly change (*p* < 0.05) and show a change of 2-fold. (**A**) Volcano plot of RNA-sequencing results showing log2 fold-change (compound/DMSO) versus statistical significance. (**B**) Scatter plot comparing RNA-sequencing data (Y-axis) versus ribosome profiling data (X-axis), both as log2 fold-change values. Colors indicate changes in mRNA levels only (green), ribosome occupancy only (red), or both (black). Pie chart shows distribution of affected genes across these categories.

**Figure 8 ijms-26-08139-f008:**
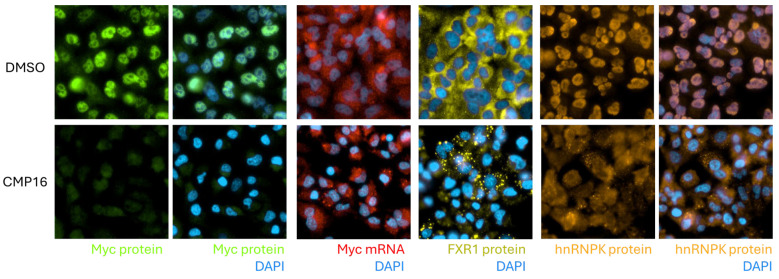
Compound treatment induces hnRNPK relocalization from nucleus to cytoplasm. A549 cells treated with CMP16 (10 μM) for 24 h, fixed, and stained with DAPI (blue), anti-Myc (green), anti-FXR1 (yellow), and anti-hnRNPK (orange) antibodies. *MYC* mRNA detected using specific FISH probes (red). Upper panels: DMSO control; lower panels: compound treatment, showing hnRNPK relocalization and stress granule formation.

**Figure 9 ijms-26-08139-f009:**
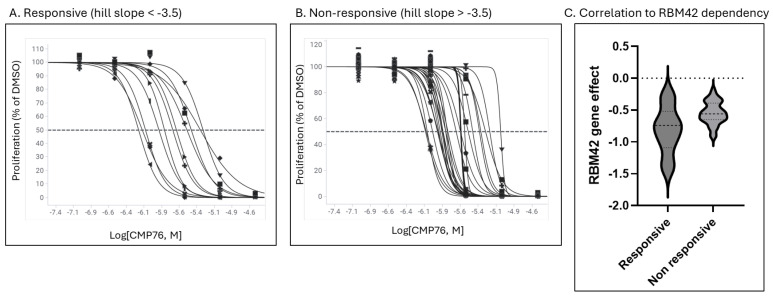
Cancer cell panel analysis reveals RBM42 dependency correlation and tissue-specific sensitivity patterns. (**A**,**B**) Panel of 43 cancer cell lines were treated with increasing compound concentrations for 72 h and EC50 values calculated from dose–response curves. Cells were divided into two groups, responsive versus non-responsive, by their EC50 values and hill slope. Each curve represents the response of a distinct cell line, with different symbols (squares, circles, triangles, etc.) used to indicate individual data points corresponding to each cell line. (**C**) Cell line dependency on RBM42 (CRISPR screen [17,18]) is plotted per group. Welch’s *T*-test *p*-value between groups = 0.0432.

## Data Availability

Compound-related OMICs data presented in this work are available upon request.

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
