# Peer review of "Discovery of Small Molecules That Inhibit MYC mRNA Translation Through hnRNPK and Induction of Stress Granule-Mediated mRNA Relocalization"

_ijms, 2025, doi:10.3390/ijms26178139_

Round 1
Reviewer 1 Report
Comments and Suggestions for Authors
This manuscript by Sheinberger et al is a novel and clever approach to use small molecules to relocalize oncogenic MYC mRNA to stress granules (SGs), a mechanism as authors point out also used by tumor cells to relocate tumor suppressor genes to SGs and promote tumor survival. The subject of the paper is unique, and the manuscript is written in a manner that is easily comprehensible. The authors have done an in-depth study by exploring multiple pathways to the effect of the small-molecule compound, assisted by some innovative techniques such as TL. I only have two minor comments-
- Some of the stress granules in confocal images are really hard to see. For example, Figure 3B. Although I am sure authors would have done an exhaustive analysis of SGs as they did provide a quantitative graph of the same, but it would be good if they could also provide better quality zoomed in images of G3BP1 marked SGs.
- Can authors elaborate a bit more on the mechanism of how the MYC mRNA translation is inhibited once they are inside condensates. Although authors have alluded it to hnRNPK activity, if authors could go into little more detail about this, it would be useful. Are the SGs blocking translational machinery to reach to the trapped mRNA?
Author Response
Thank you for taking the time to review this manuscript. Please find detailed responses below and the corresponding revisions/corrections in track changes in the re-submitted files.
|
|
Yes |
Can be improved |
Must be improved |
Authors comment |
|
Does the introduction provide sufficient background and include all relevant references? |
(x) |
( ) |
( ) |
|
|
Is the research design appropriate? |
(x) |
( ) |
( ) |
|
|
Are the methods adequately described? |
(x) |
( ) |
( ) |
|
|
Are the results clearly presented? |
(x) |
( ) |
( ) |
|
|
Are the conclusions supported by the results? |
(x) |
( ) |
( ) |
|
|
Are all figures and tables clear and well-presented? |
( ) |
(x) |
( ) |
See response to comment 1 |
Comment 1: Some of the stress granules in confocal images are really hard to see. For example, Figure 3B. Although I am sure authors would have done an exhaustive analysis of SGs as they did provide a quantitative graph of the same, but it would be good if they could also provide better quality zoomed in images of G3BP1 marked SGs.
Response: Thank you for pointing this out, indeed it is important to show the phenotype of the SGs, and larger images can show this. We have prepared figure with zoomed in images to show SGs. This figure will be added as supplementary figure 2 (article, lines 813-820). Please see the figure and its legend below. In addition, please see under “track changes”, reference to this figure in the article (rows 183-186).
Comment 2: Can authors elaborate a bit more on the mechanism of how the MYC mRNA translation is inhibited once they are inside condensates. Although authors have alluded it to hnRNPK activity, if authors could go into little more detail about this, it would be useful. Are the SGs blocking translational machinery to reach to the trapped mRNA?
Response: We agree that the original discussion did not fully detail how stress granules function as translationally repressive compartments or explain why ribosomes cannot access the sequestered mRNA. To address this, we have added a new paragraph following the RBM42/hnRNPK paragrpah that elaborates on the link between SG sequestration and translation inhibition (see lines 408-422, track changes). This addition clarifies the spatial and molecular mechanisms underlying MYC mRNA repression and strengthens the connection between our study’s mechanistic aims and its conclusions. We appreciate the reviewer’s comment, which has helped us improve the clarity and completeness of the discussion.
Reviewer 2 Report
Comments and Suggestions for Authors
Despite the relatively clear presentation of the data obtained, the manuscript contains several serious shortcomings that prevent its acceptance for publication.
Based on the title of the manuscript and the abstract, I expected its content to be completely different.
The authors present a large amount of data that cannot be verified and may not be true. The corresponding author did not even bother to provide his full mailing address. The authors claim to have analyzed a huge number of small molecules that could selectively modulate MYC translation. This claim is not verifiable and is not supported in any way in the manuscript. In addition, the reader has no idea what the molecules listed under the abbreviations CMP11, 16, and 76 represent; their structure is not given, nor is any reference provided.
A scientific article containing a number of unconfirmed and incomplete facts is not credible.
Author Response
Thank you for taking the time to review this manuscript. Please find detailed responses below and the corresponding revisions/corrections in track changes in the re-submitted files.
|
|
Yes |
Can be improved |
Must be improved |
Authors comment |
|
Does the introduction provide sufficient background and include all relevant references? |
(x) |
( ) |
( ) |
|
|
Is the research design appropriate? |
( ) |
( ) |
(x ) |
See response below |
|
Are the methods adequately described? |
( ) |
( ) |
(x ) |
See response below |
|
Are the results clearly presented? |
( ) |
( ) |
(x ) |
See response below |
|
Are the conclusions supported by the results? |
( ) |
( ) |
(x ) |
See response below |
|
Are all figures and tables clear and well-presented? |
( ) |
(x) |
( ) |
See response below |
Comment 1: Based on the title of the manuscript and the abstract, I expected its content to be completely different.
Response: The article showcases the discovery, in a step-wise manner of small molecules which regulate MYC mRNA translation by sequestering it to translation silent RNA granules; We start with a description of of development and validation of a MYC mRNA translation assay, TranslationLight, and its’ use in a high throughput screen using a diverse library. It follows with evidence of reduction of Myc protein and an induction of relocalization of Myc mRNA. Next, we show how we arrived at the hypothesis that the compounds induce stress granules (SGs), and what molecular steps occur prior to SG formation. Finally, we show how a target was identified for the compound, an RBP implicated in MYC mRNA regulation, hnRNPK. We also attempted to identify cancer cells that are sensitive to the compound by using a panel of tumor cells. The abstract describes, step-by-step, the data presented in the results section. Thus, we believe that the title, “Discovery of Small Molecules that Inhibit MYC mRNA Translation through hnRNPK and induction of Stress Granule-Mediated mRNA Relocalization” and abstract correctly reflects the content of the manuscript.
Comment 2: The authors present a large amount of data that cannot be verified and may not be true.
Response: Cell images, western blots are presented with the data that is extracted from them and presented as graphs. The OMICs data is available upon request also for reviewers. The only images that are not shown are the ones for the screen itself; however, assay validation images and data are presented. Data for the screen is presented with statistical analysis. The validation that the compounds reduce Myc protein but does not affect its mRNA supports the screen results.
Comment 3: The corresponding author did not even bother to provide his full mailing address.
Response: the corresponding author (She) included the email address in the preprint, as instructed by the publisher. We added the full address in track changes (line 8).
Comment 4: The authors claim to have analyzed a huge number of small molecules that could selectively modulate MYC translation. This claim is not verifiable and is not supported in any way in the manuscript.
Response: We have used TranslationLight (previously known as Protein Synthesis Monitoring), a peer reviewed and published methodology that monitors mRNA translation (see references at the end of this document). We have conducted an orthogonal study to TranslationLight, puromycilation, and shown that the compounds inhibit incorporation of puromycin to newly synthesized Myc, to a similar extent as translation inhibitors, homoharringtonine and cycloheximide (supplementary Figure 1, lines 804-812 in the manuscript). We have added in the results section a reference to this figure and added a description of the method to the method section (lines 146-150).
We thank the reviewer for noticing an omission of the library source and size from the methods section. We added this information in lines 462-463).
Comment 5: In addition, the reader has no idea what the molecules listed under the abbreviations CMP11, 16, and 76 represent; their structure is not given, nor is any reference provided.
Response: Compound structures were added below the graph in Figure 1E. A reference was added to the patent claiming the molecules (reference 11 in the article).
TranslationLight References
- Koltun B, Ironi S, Gershoni-Emek N, Barrera I, Hleihil M, Nanguneri S, Sasmal R, Agasti SS, Nair D, Rosenblum K. Measuring mRNA translation in neuronal processes and somata by tRNA-FRET. Nucleic Acids Res. 2020 Apr 6;48(6):e32. doi: 10.1093/nar/gkaa042. PMID: 31974573; PMCID: PMC7102941.
- Plochowietz A, Farrell I, Smilansky Z, Cooperman BS, Kapanidis AN. In vivo single-RNA tracking shows that most tRNA diffuses freely in live bacteria. Nucleic Acids Res. 2017 Jan 25;45(2):926-937. doi: 10.1093/nar/gkw787. Epub 2016 Sep 12. PMID: 27625389; PMCID: PMC5314786.
- Liu J, Pampillo M, Guo F, Liu S, Cooperman BS, Farrell I, Dahary D, Gan BS, O'Gorman DB, Smilansky Z, Babwah AV, Leask A. Monitoring collagen synthesis in fibroblasts using fluorescently labeled tRNA pairs. J Cell Physiol. 2014 Sep;229(9):1121-9. doi: 10.1002/jcp.24630. PMID: 24676899.
- Barhoom S, Farrell I, Shai B, Dahary D, Cooperman BS, Smilansky Z, Elroy-Stein O, Ehrlich M. Dicodon monitoring of protein synthesis (DiCoMPS) reveals levels of synthesis of a viral protein in single cells. Nucleic Acids Res. 2013 Oct;41(18):e177. doi: 10.1093/nar/gkt686. Epub 2013 Aug 21. PMID: 23965304;PMCID: PMC3794613.
- Rosenblum G, Chen C, Kaur J, Cui X, Zhang H, Asahara H, Chong S, Smilansky Z, Goldman YE, Cooperman BS. Quantifying elongation rhythm during full-length protein synthesis. J Am Chem Soc. 2013 Jul 31;135(30):11322-9. doi:10.1021/ja405205c. Epub 2013 Jul 16. PMID: 23822614; PMCID: PMC3768011.
- Stevens B, Chen C, Farrell I, Zhang H, Kaur J, Broitman SL, Smilansky Z, Cooperman BS, Goldman YE. FRET-based identification of mRNAs undergoing translation. PLoS One. 2012;7(5):e38344. doi: 10.1371/journal.pone.0038344. Epub 2012 May 31. PMID: 22693619; PMCID: PMC3365013.
- Chen C, Stevens B, Kaur J, Smilansky Z, Cooperman BS, Goldman YE. Allosteric vs. spontaneous exit-site (E-site) tRNA dissociation early in protein synthesis. Proc Natl Acad Sci U S A. 2011 Oct 11;108(41):16980-5. doi:10.1073/pnas.1106999108. Epub 2011 Oct 3. PMID: 21969541; PMCID: PMC3193197.
- Barhoom S, Kaur J, Cooperman BS, Smorodinsky NI, Smilansky Z, Ehrlich M, Elroy-Stein O. Quantitative single cell monitoring of protein synthesis at subcellular resolution using fluorescently labeled tRNA. Nucleic Acids Res. 2011 Oct;39(19):e129. doi: 10.1093/nar/gkr601. Epub 2011 Jul 27. PMID: 21795382; PMCID: PMC3201886.
- Chen C, Stevens B, Kaur J, Cabral D, Liu H, Wang Y, Zhang H, Rosenblum G, Smilansky Z, Goldman YE, Cooperman BS. Single-molecule fluorescence measurements of ribosomal translocation dynamics. Mol Cell. 2011 May 6;42(3):367-77. doi:10.1016/j.molcel.2011.03.024. PMID: 21549313; PMCID: PMC3090999.
- Bharill S, Chen C, Stevens B, Kaur J, Smilansky Z, Mandecki W, Gryczynski I, Gryczynski Z, Cooperman BS, Goldman YE. Enhancement of single-molecule fluorescence signals by colloidal silver nanoparticles in studies of protein translation. ACS Nano. 2011 Jan 25;5(1):399-407. doi: 10.1021/nn101839t. Epub 2010 Dec 16. PMID: 21158483; PMCID: PMC3049198.
Round 2
Reviewer 2 Report
Comments and Suggestions for Authors
The revised manuscript is now acceptable; only a few minor modifications and additions of key information were necessary.